# The origin of suspended particulate matter in the Great Barrier Reef

Mohammad Bahadori [1,2], Chengrong Chen [1,2] ✉, Stephen Lewis [3], Juntao Wang [4,5], Jupei Shen[6], Enqing Hou[7], Mehran Rezaei Rashti [1,2], Qiaoyun Huang [8], Zoe Bainbridge[3] & Tom Stevens[3]

River run-off has long been regarded as the largest source of organic-rich suspended particulate matter (SPM) in the Great Barrier Reef (GBR), contributing to high turbidity, pollutant exposure and increasing vulnerability of coral reef to climate change. However, the terrestrial versus marine origin of the SPM in the GBR is uncertain. Here we provide multiple lines of evidence ($^{13}$C NMR, isotopic and genetic fingerprints) to unravel that a considerable proportion of the terrestrially-derived SPM is degraded in the riverine and estuarine mixing zones before it is transported further offshore. The fingerprints of SPM in the marine environment were completely different from those of terrestrial origin but more consistent with that formed by marine phytoplankton. This result indicates that the SPM in the GBR may not have terrestrial origin but produced locally in the marine environment, which has significant implications on developing better-targeted management practices for improving water quality in the GBR.

The Great Barrier Reef (GBR) is the world's largest coral reef ecosystem, stretching over 2,000 km along the coast of Queensland, Australia (Fig. 1). Currently, the condition of GBR is deteriorating due to a combination of global (e.g., climate change) and local (e.g., terrestrial runoff) stressors[1]. Discharges from rivers can degrade water quality by carrying pollutants including nutrients and sediment into the inshore waters of the GBR[2]. Simulation modelling as well as monitoring has shown that a bulk mass of catchment sediments in the GBR region deposits within a short distance from river mouths[3,4]. However, a very fine fraction of sediments has the capacity to transform into organic-rich suspended particulate matter (SPM: particulate organic matter and mineral sediment) and be transported further offshore over long distances[4]. This organic-rich SPM is the most detrimental form of sediment which is easily resuspended and can have a prolonged effect on water clarity[5] and reduce the availability of

photosynthetically usable light for benthic phototrophs such as coral reefs and seagrass meadows[6,7].

To mitigate the impact of water quality in the GBR, the Australian and Queensland Governments have developed a long-term plan, the Reef 2050 Water Quality Improvement Plan (WQIP)[8]. This plan specifically targets identifying and mitigating the land-based sourced pollution to the GBR, with the SPM being one of the greatest concerns[9]. While there is little doubt about the terrestrial origin of the SPM mineral component[10,11], the terrestrial versus marine origin of the SPM organic component in the GBR is largely unknown. Plants provide the dominant source of organic matter (OM) in the terrestrial environment, and rivers are predominantly responsible for the export of terrestrially-derived OM to river estuaries and subsequently to the marine environment[12]. River estuaries are highly dynamic environments in which heterotrophy (OM decomposition) generally dominates autotrophy (OM production)[13] and the SPM organic component

[1]Australian Rivers Institute, Griffith University, Nathan, QLD 4111, Australia. [2]School of Environment and Science, Griffith University, Nathan, QLD 4111, Australia. [3]Catchment to Reef Research Group, Centre for Tropical Water and Aquatic Ecosystem Research, James Cook University, Townsville, QLD, Australia. [4]Hawkesbury Institute for the Environment, Western Sydney University, Penrith, NSW, Australia. [5]Global Centre for Land-Based Innovation, Western Sydney University, Penrith, NSW, Australia. [6]School of Geographical Sciences, Fujian Normal University, Fuzhou, PR China. [7]Key Laboratory of Vegetation Restoration and Management of Degraded Ecosystems, South China Botanical Garden, Chinese Academy of Sciences, Guangzhou 510650, China. [8]State Key Laboratory of Agricultural Microbiology, Huazhong Agricultural University, Wuhan, China. ✉e-mail: c.chen@griffith.edu.au

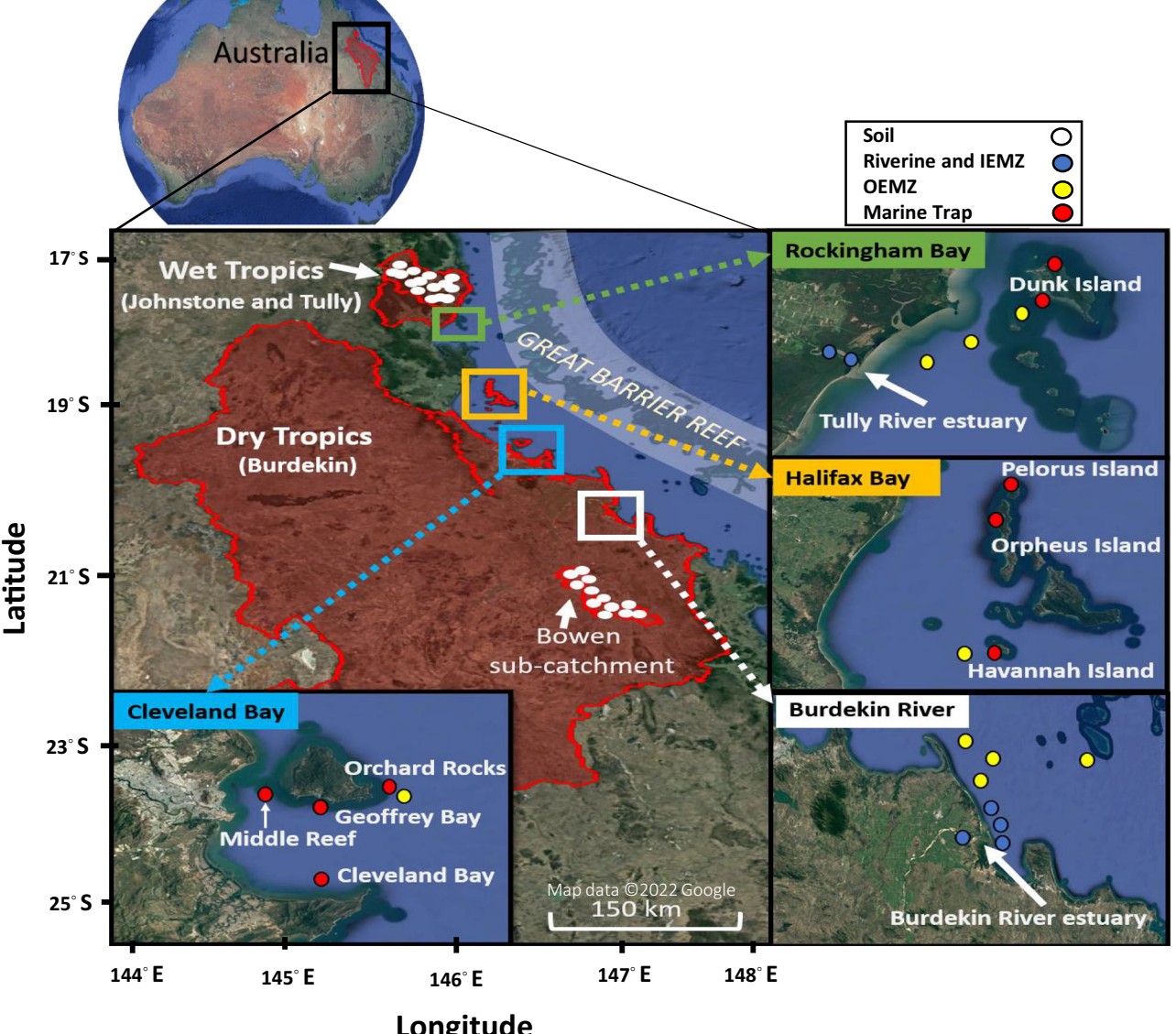

**Fig. 1 | Sampling locations in the catchments and lagoon of the Great Barrier Reef, north Queensland, Australia.** Collected samples include soil, suspended particulate matter (SPM) in riverine and inner estuarine mixing zone (IEMZ); SPM collected from outer estuarine mixing zone (OEMZ) and marine trap sediment (Map data ©2022 Google).

may be partly or entirely mineralised[14] before reaching the marine environment. The extent of SPM mineralisation is controlled by the chemical composition of the organic component (related to OC composition), the biological capability and capacity of associated microbial communities (biological component), and the physical mechanisms of protection provided by mineral sediment (mineral component) of SPM[15]. In addition to terrestrial inputs, the SPM organic component could also be produced locally within the estuarine and marine environment[16]. Phytoplankton can produce OM by taking up inorganic C and nutrients through photosynthesis in the sunlit ocean[17]. There is also evidence of autotrophic bacteria capable of synthesising organic C (OC) by the reduction of $CO_2$ in the marine environment[18].

In estuarine and inshore marine environments, a combination of degradative and synthetic processes governs the relative contribution of terrestrial and marine sources to the SPM organic components. Degradation of OM to small, soluble molecules and release to solution during biological metabolism and oxidative respiration are the primary modes of OM loss, while the autotrophic and heterotrophic production of biomass represent major biosynthetic pathways that

occur simultaneously with OM degradation[19]. Thus, SPM organic components are derived from a variety of terrestrial and marine origins, however, the relative contribution of these sources across the freshwater-estuarine-marine continuum of the GBR is uncertain. We argue that this uncertainty is symptomatic of an emerging paradigm that, until now, has not adequately recognised the changes in mineral, organic and biological components of the SPM in the mediums (freshwater-estuarine-marine) in which it is transported from catchment to reef. Indeed, in such dynamic mediums, the mineral (e.g., particle size distribution), and biological (bacterial and fungal community composition) components of SPM are also in a continuous state of change, with concomitant effects on the SPM organic components.

Here, we specifically aimed to unravel the terrestrial versus marine origins of SPM organic components in the GBR lagoon by incorporating multiple lines of evidence (structural, isotopic and genetic fingerprints) into a comprehensive conceptual model that links the structural components of SPM and their interrelated behavioural aspects during transportation from catchment to reef. It was

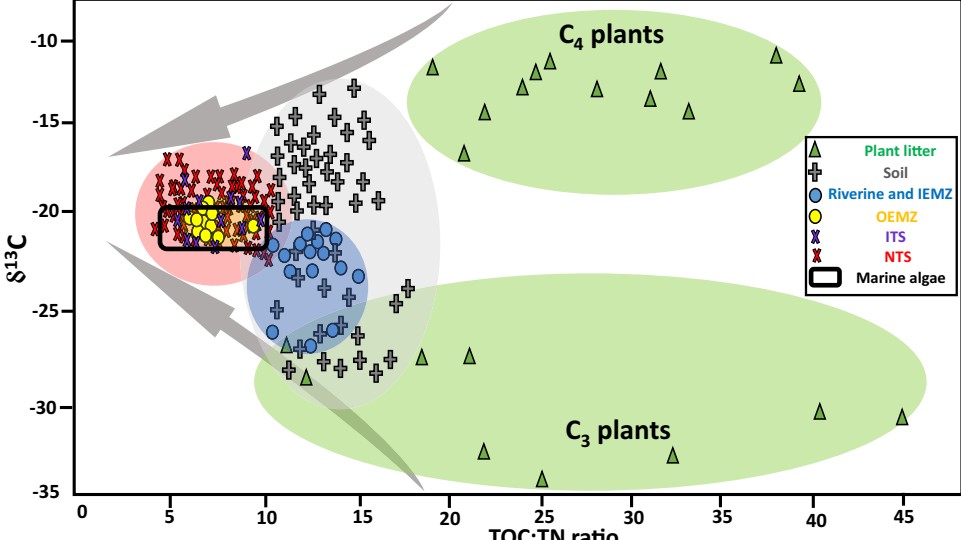

**Fig. 2 | The origin of suspended particulate matter (SPM) organic component-stable isotope fingerprint.** Terrestrial versus marine sources of organic matter and their converging pathway along the terrestrial plant-soil-riverine-estuarine-marine continuum of the Great Barrier Reef (GBR). Total organic carbon to total nitrogen ratio (TOC:TN) and stable isotope carbon ($\delta^{13}$C) values are presented for plant litter, soil, SPM in riverine and inner estuarine mixing zone (IEMZ); SPM collected from outer estuarine mixing zone (OEMZ); sediment collected by marine traps influenced by flood plume (ITS) and sediment collected by marine traps not influenced by flood plume (NTS). Identifier of OM produced by marine algae is from previous studies[20–22]. Ellipses highlight the distribution of samples in each category.

hypothesised that the SPM organic components in the GBR lagoon may not be of terrestrial origin. Marine-derived OM could form a substantial part of the organic-rich SPM offshore, while the majority of terrestrially-derived OM would be decomposed within the riverine and estuarine mixing zones.

## Results and discussion

### Origin of SPM organic component-Stable isotope fingerprint

The $\delta^{13}$C and total organic carbon to total nitrogen (TOC:TN) ratios of the SPM organic component changed considerably during transportation across terrestrial to marine environments of the GBR and these changes were particularly evident in the estuarine mixing zones (Fig. 2). The SPM in riverine and inner estuarine mixing zone (IEMZ) samples had $\delta^{13}$C values of −21 to −27‰ and TOC:TN ratios of 10–15 which were within the ranges measured for soils ($\delta^{13}$C −13 to −28‰ and TOC:TN ratio 10–18) across the Wet Tropics and Dry Tropics regions. As SPM passed from riverine and IEMZ into the outer estuarine mixing zone (OEMZ), the characteristics of its organic component ($\delta^{13}$C −19 to −21‰ and TOC:TN ratio 6–10) became more consistent with that produced by marine algae[20–22]. Moreover, the $\delta^{13}$C and TOC:TN ratios of SPM collected from OEMZ fell within the ranges measured in the marine trap samples not influenced by flood plumes (Fig. 2).

Such decreases in TOC:TN ratios of the SPM organic component in the OEMZ are indicative of fast-growing marine phytoplankton cells and reflect increases in the concentrations of both proteins and nucleic acids relative to vascular plant tissues. Indeed, proteins and peptides are the major fractions of N-containing molecules in living organisms and are expected to form a significant part of marine-derived OM[23]. Moreover, marine organisms do not require the C-rich polymeric materials (e.g., cellulose and lignin) that terrestrial plants usually produce to physically support their structure. Therefore, the predominance of proteins (TOC:TN ratio-3–4) compared to N-free bio-macromolecules (e.g., cellulose) can distinguish the marine-derived OM (TOC:TN ratio-7) from characteristically C-rich (TOC:TN ratio 10–45) plant tissues (Fig. 2).

The stable $\delta^{13}$C isotope is another discriminative tool for identifying the terrestrial versus marine source of SPM organic components as photosynthetic species in terrestrial and marine environments use different sources of inorganic C to produce OM. Bicarbonate ($\delta^{13}$C-0‰) is the main source of inorganic C for marine algae[20,21] while terrestrial plants consume atmospheric $CO_2$ with an average $\delta^{13}$C value of −−8.4‰[24]. Marine-derived OM typically have $\delta^{13}$C values within a narrow range of −20 to −22‰[20,21], while the OM synthesised by terrestrial plants have a much wider range of isotopic $\delta^{13}$C values depending on their photosynthesis system ($C_4$ or $C_3$). Specifically, in this study the high variation of $\delta^{13}$C in soils is due to the types of vegetation which serve as the source of OM within and between catchments of the GBR. Forest ($C_3$ plants with a mean $\delta^{13}$C of −30‰) is the dominant vegetation type in the Wet Tropics region, while the Burdekin catchment (Dry Tropics) is dominated by grassland ($C_4$ plants with a mean $\delta^{13}$C of −13‰). Despite the high variation in terrestrial plants, when the shifts in $\delta^{13}$C and TOC:TN ratios of the SPM organic component across the terrestrial and marine samples were considered, it was apparent that the direction of changes converged towards the marine environment, with $\delta^{13}$C values and TOC:TN ratios consistent with the formation of OM similar to marine algae (Fig. 2).

### Structure of SPM organic component-13C NMR fingerprint

The OM produced by terrestrial plants had a high proportion of O-alkyl C (−64%, representing mainly polysaccharides) that decreased considerably (down to 34%) during transportation along the terrestrial-freshwater-estuarine-marine continuum, while the proportion of alkyl C (representing mainly aliphatic compounds) significantly increased from 14% to 33% (Fig. 3). "Lability" is a term to describe the decomposition rate of organic compounds[25], such that "labile" compounds (e.g., polysaccharides such as cellulose, hemicellulose) normally have short residence times in the environment, while "recalcitrant" compounds (e.g., lipid/aliphatic compounds such as fatty acids, waxes, cutin, suberin) are relatively slowly degraded or resistant to further degradation by microorganisms. Heterotrophic microorganisms consume OM to synthesise their own tissues and metabolic products during the process of decomposition[26]. These microorganisms first decompose the more labile fractions of OM (e.g., O-alkyl C), and as decomposition proceeds, compounds with more recalcitrant chemical structures (e.g., alkyl C) tend to accumulate. The selective decomposition of labile compounds and preservation of more recalcitrant

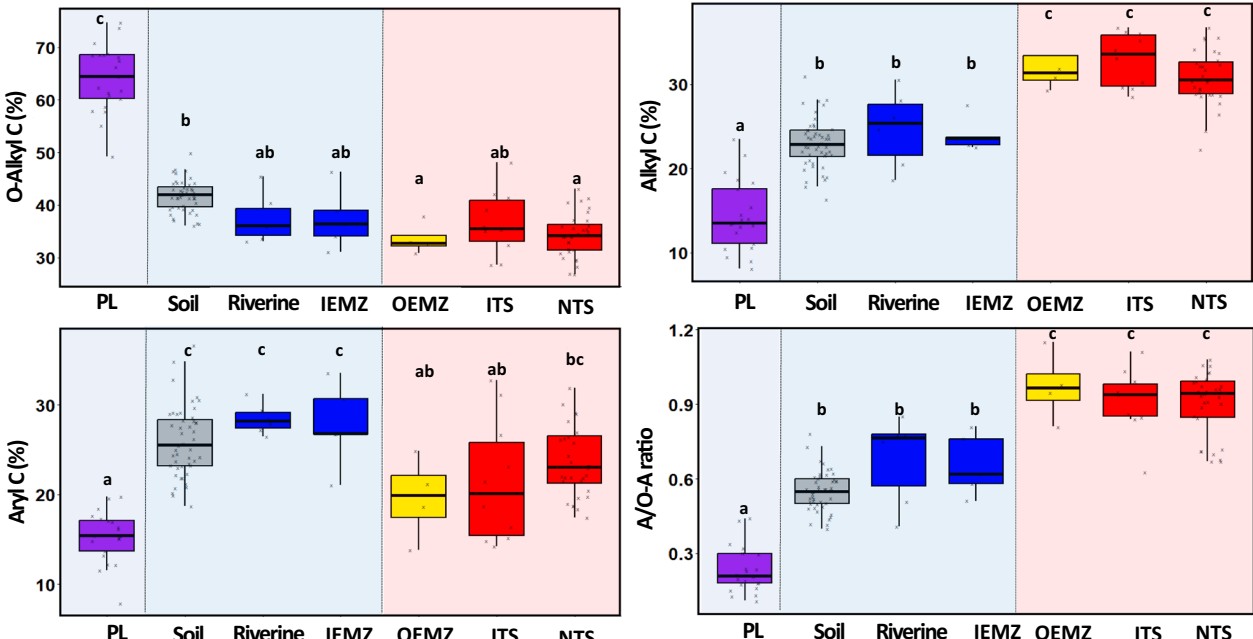

**Fig. 3 | The structure of suspended particulate matter (SPM) organic component–¹³C NMR fingerprint.** Changes in the proportion of organic carbon functional groups (O-alkyl C, alkyl C, aryl C) and the ratio of alkyl C to O–alkyl C functional groups (A/O–A ratio) during the transportation of SPM along the terrestrial plant-soil-riverine-estuarine-marine continuum of the Great Barrier Reef. PL: terrestrial plant litter; Riverine: the SPM collected from river; IEMZ: the SPM collected from inner estuarine mixing zone; OEMZ: the SPM collected from outer estuarine mixing zone; ITS: marine trap sediment influenced by flood plumes; NTS: marine trap sediment not influenced by flood plumes. Box plots with different letter (e.g., a, b, c) are significantly different at $p < 0.05$ (error bars represent SD).

forms of OC was also evident in the significant increase in the alkyl C to O-alkyl C ratios of OM (Fig. 3) during transportation from the terrestrial (-0.2) to the marine environment (-1.0).

The shift in the OM chemical structure was particularly notable within the estuarine mixing zones where the proportion of aryl C significantly decreased from the IEMZ to the OEMZ. The aromatic biomolecules containing aryl C have lower lability compared to polysaccharides[27]. Lignin, for example, is a heterogeneous aromatic (phenolic) biomacromolecule that is the second most abundant polymer in nature after cellulose[28]. The primary role of lignin is to confer structural strength to the terrestrial plant tissue and rigidity to the cell wall[29]. Thus, the significant decline in the proportion of aryl C within estuarine zones was because lignin-derived compounds are essentially absent in marine-derived OM but significantly contribute to terrestrially-derived OM. In addition, the shift in environmental conditions throughout the estuarine zones favoured the dominance of bacteria over fungi and changed the bacterial community composition and structure (Fig. 4c–e). Such shifts in microbial communities are toward a better biological capability and capacity in decomposing the OM that enter the marine environment[15]. Proteobacteria and Actinobacteria were the dominant bacterial groups in soil, riverine and IEMZ while Proteobacteria and Bacteroidetes were the highest in the marine environment including OEMZ and marine trap sediment (Fig. S6). Specific prokaryotic functional taxa (e.g., Proteobacteria) and their lignin degradation functions have been identified as potential polymeric lignin or its aromatic fragments degraders in the marine environment[30]. Thus, terrestrially-derived OM with high lignin contents can be further decomposed within a short distance from river estuary (OEMZ) by some marine bacteria that possess enzymes, such as ring-opening dioxygenases, to catabolize aromatic compounds[31,32]. From these observations, we derived the hypothesis that a considerable proportion of the SPM organic component is decomposed during the transportation within riverine and estuarine mixing zones, consequently becoming more resistant to further rapid C loss in the marine environment. The remaining recalcitrant biomacromolecules

represent a small fraction of the initial biomass produced by terrestrial plants and are likely to make up a major fraction of the OM that eventually is deposited further offshore.

## Microbial community composition and structure-Genetic fingerprint

Structural equation modelling (SEM) was used to provide additional insights on the complex interrelationships among the SPM organic, mineral and biological components across soil-riverine-estuarine-marine habitats (Fig. 4a). The biological component of SPM represented by microbial diversity and community structure (MDCS), consisting of bacterial and fungal richness (BactRich and FunRich), diversity (Shannon Index - BactShann and FunShann) and community structure (BactNMDS and FunNMDS)[33,34]. Our results showed that the shift in MDCS was highly correlated with the variation in environmental factors (Env2) and organic carbon characteristics (OCC) of the SPM across habitat types (Fig. 4a). Network analysis indicated that the shift in microbial community composition across habitats was particularly influenced by the variation in environmental factors such as the characteristics of mineral component (e.g., particle size distribution) and solution chemistry (e.g., salinity, pH) of the mediums in which SPM was transported along the soil-riverine-estuarine-marine continuum (Fig. 4b). This result indicates the role of ecological factors in regulating microbial diversity and community structure across habitats (Fig. 4d, e). The higher bacterial diversity observed in soil and riverine habitats was attributed to the multiplicity of ecological niches and greater habitat variations compared to the marine environment[35]. Habitat variability is created by physicochemical gradients, nutrient concentrations, variation of organic substrate, plant diversity and interactions between microorganisms at different trophic levels[36]. Such variability is more likely to occur in soils of different types under different vegetations, and also riverine habitats compared to more homogeneous habitats such as the marine environment. Given that only SPM with very fine particle size is able to reach further offshore, a dramatic decrease in the concentration of SPM is expected moving

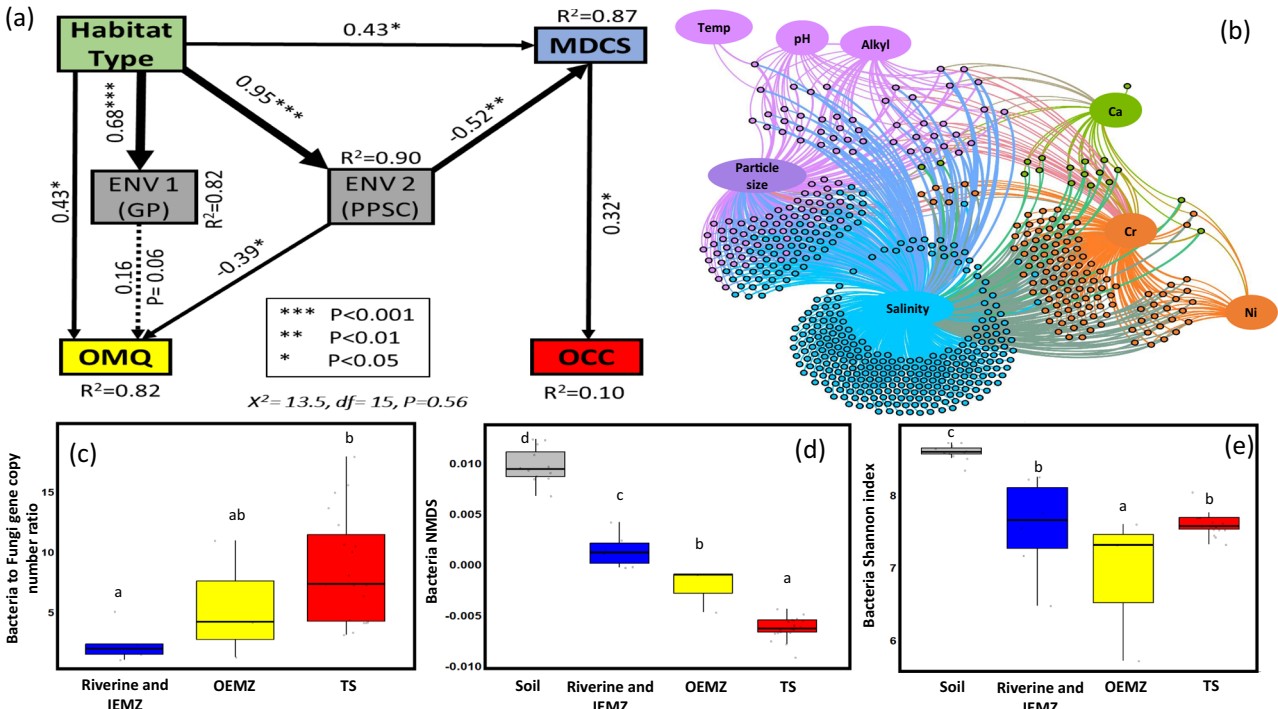

**Fig. 4 | Microbial community composition and structure-Genetic fingerprint.**
This figure represents **a** structural equation model (SEM) to assess the effects of habitat type (soil, riverine, estuarine and marine habitats), environmental factors (Env1 and Env2) and microbial diversity and community structure (MDCS) on organic matter quality (OMQ) and organic carbon characteristics (OCC) across terrestrial and marine environments. The OMQ included total organic carbon (TOC), TOC to total nitrogen ratio, $\delta^{13}C$, $\delta^{15}N$, $^{13}C$ NMR functional groups (carboxyl C, aryl C, O-alkyl C and alkyl C). The $^{13}C$ NMR functional groups aryl C, O-alkyl C and alkyl C were separately grouped as OCC of terrestrial and marine habitats. Environmental factors were divided into two groups including geochemical properties (ENV1-GP) and physical properties and solution chemistry (ENV2-PPSC). The ENV1-GP included Mn, Al, Ni, Co, Cr, Fe and P. The ENV2-PPSC included pH, salinity, Na, K, Ca, Mg, temperature, and particle size fraction <20 μm of soil and suspended particulate matter (SPM). The MDCS was included in the SEM model to represent

the shift in bacterial and fungal richness (BactRich and FunRich), diversity (Shannon Index- BactShann and FunShann) and community structure (BactNMDS and FunNMDS) across habitats including soil and SPM collected from riverine and inner estuarine mixing zone (IEMZ), outer estuarine mixing zone (OEMZ) and marine trap sediment (TS). In the SEM diagram, the numbers on arrows are standardised path coefficients. Arrow width is proportional to the significance level of the standardised coefficient. The dashed arrow indicates a marginal path with p-value = 0.06. $R^2$ indicates the proportion of the variance explained for each dependent variable in the model; **b** network analysis of microbial OTUS and soil and SPM characteristics (as described above); **c** changes in bacteria to fungi gene copy number ratios; **d** shift in bacterial community structure (BactNMDS); and **e** bacterial diversity (Shannon Index) across habitats. Box plots with the same letter (e.g., a, b, c) are not significantly different at $p < 0.05$ (error bars represent SD).

towards OEMZ and marine environments[37]. Thus, access to surface area and primary resources on SPM is limited towards the marine environment causing stronger competition among bacterial groups[38]. Limited resources and stronger trophic interactions among microbial groups can be a powerful strategy to deter competitors on SPM[39,40]. Thus, such a transportation pathway (soil-riverine-estuarine-marine) moves towards a microbial community structure in the marine environment which is significantly different from terrestrial origin (Fig. 4d). Combined, these results suggested that the SPM biological component is in a continuous state of change due to the variation in the physiochemical properties of the medium (soil-riverine-estuarine-marine) in which the SPM is transported from catchment to reef.

## Conceptual model and implications

We have developed a conceptual model to provide insights into the origin and fate of organic-rich SPM along the continuum spanning from soil and riverine to estuarine and marine environments of the GBR (Fig. 5). According to this model, most of the terrestrially-derived OM is either physically deposited near river mouth or decomposed within the estuarine and marine environments by numerous biotic (biological degradation) and abiotic (photochemical oxidation) processes in response to the strong physiochemical and biological gradients. In this study sediment deposited near the river mouths (within the IEMZ) of the flood plume has not been sampled during and after

flood periods to examine how the terrestrial OM is degraded over time. However, previous studies on the processing of terrestrial OM in river estuaries of the GBR have documented the rapid microbial degradation in such locations including high tidally driven estuarine zones[41,42]. According to our detailed fingerprinting analysis, there is a rapid change in the isotopic, structural and genetic fingerprints of the SPM between the IEMZ and the OEMZ, which highlights that terrestrial OM inputs are almost exclusively deposited near river mouths with very little (if any) even reaching our inshore monitoring sites. In addition to the loss/transformation of terrestrially-derived OM, the contribution of primary production (phytoplanktonic photosynthesis) to SPM organic component increases towards the marine environment. These changes are due to the lower terrestrial OC supply and better light conditions in which phytoplankton may draw their nutrient requirements through dissolved nutrients from both terrestrial sources in flood plume and local marine mineralisation[43]. Together, these processes change the isotopic, structural and genetic fingerprints of the SPM leaving estuaries making it completely different from those of terrestrial origin but more consistent with that formed locally by marine phytoplankton. Other source of productivity in the marine environment includes *Trichodesmium* blooms which can fix nitrogen and form marine OM further offshore[44]. Indeed, marine OM associated with *Trichodesmium* needs further characterisation which is pertinent with the purported increases in the GBR lagoon in recent times[45].

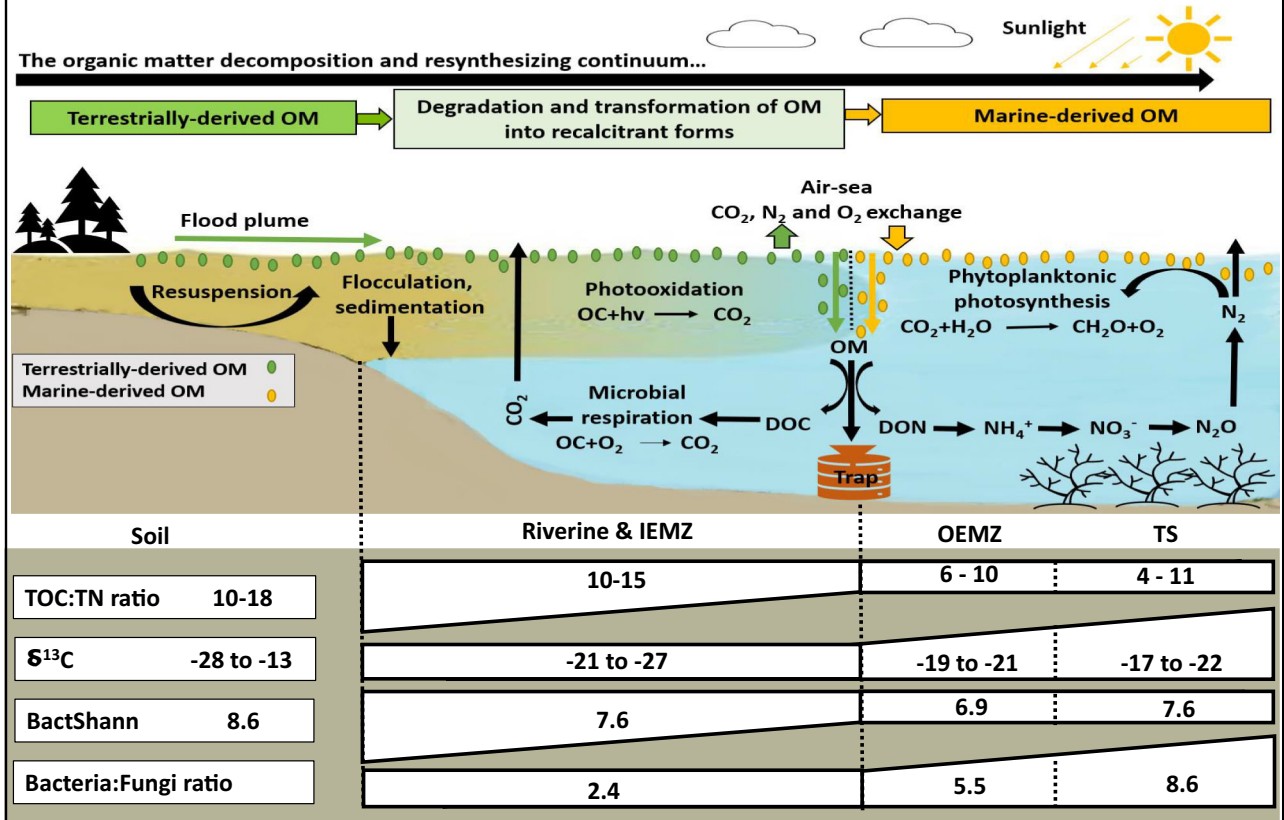

**Fig. 5 | Schematic diagram of the origin of suspended particulate matter-organic matter (OM) component during transportation from terrestrial to the marine environment.** The major processes related to the organic carbon (OC) and nitrogen (N) cycles, changes in the stable isotope carbon ($\delta^{13}$C) value, total OC to total N ratio (TOC:TN ratio), bacterial diversity (Shannon index - BactShann) and bacteria to fungi gene copy number ratio (Bacteria:Fungi ratio) are illustrated for OM and the associated microbial community during transportation along the soil, riverine and inner estuarine mixing zone (IEMZ) to outer estuarine mixing zone (OEMZ) and marine trap sediment (TS). DOC: dissolved organic carbon; DON: dissolved organic nitrogen.

Several studies have examined and identified the role that terrestrially-derived SPM plays in affecting rates of sedimentation and water clarity in the riverine and inner estuarine environments[3,37,46], however, further research is urgently required to identify the key environmental drivers of the production of organic-rich SPM in the outer estuarine and marine environments. This motivates demands for a new conceptual framework that adequately recognises and addresses both terrestrial and marine sources of SPM in the GBR.

## Methods
### Study area
The area adjacent to the GBR lagoon is composed of 35 river basins/catchments covering an area of approximately 423,000 km² along the eastern coast of Queensland, Australia[47]. These catchments are situated across a wide range of climates including the Wet Tropics and Dry Tropics regions, resulting in a variety of geologies and soils, geomorphologies, vegetation types, hydrological and rainfall patterns within and between catchments[48].

The Wet Tropics region comprises 5% of the total GBR catchment area, however, this region contributes a large proportion of total nitrogen loads (TN~31%) exported to the GBR lagoon[49,50]. The Johnstone and Tully catchments, collectively, cover ~19% of the Wet Tropics region (~4,532 km²) and have an average annual rainfall of ~3,500 mm (Fig. 1). The Wet Tropics region is generally characterised by wet summers and relatively dry winters, highly influenced by tropical rain depressions especially during the wet season between December and April. In terms of vegetation, a large area (~52%) of the Wet Tropics region has been set aside for conservation and forestry purposes, while other land uses include cattle and dairy (~34%), sugarcane (~9%), bananas (~2%) and other land uses[47,51].

The Burdekin catchment is situated in the Dry Tropics (catchment area ~130,120 km²) with an average rainfall of ~900 mm per year. A large proportion of the catchment (~90%) is covered by grazing pastures with some conservation forest lands (~5%)[47]. Most of the considerable catchment-wide rainfall events in the Dry Tropics that generate flooding are related to tropical cyclones and low-pressure systems between the wet season months of December and April. The vegetation of the Burdekin region includes eucalypt savannah, acacias (Brigalow Belt) and grasslands[52]. The Burdekin region is a major contributor of sediment to the GBR lagoon with approximately 40% of the total anthropogenic sediment load[50,52].

### Terrestrial plant and soil sampling
The selected catchments varied in their land use, catchment area, geographical location, vegetation type, geological structure, and climate, thereby capturing a wide range of terrestrially-derived SPM with various chemical and isotopic signatures/endmembers and microbial diversity and community structure. A total of 35 soil samples were collected to characterise the four key land uses within the Johnstone catchment (grazing pastures: 9 samples, sugarcane: 10 samples, rainforest: 8 samples and bananas: 8 samples) in July 2016[51]. For each soil sample, a composite of five separate sampling points was collected and homogenised from the surface (0–10 cm) after vegetation was removed. Homogenous subsamples of the Johnstone catchment fresh soils (three replicates from each land use) were then frozen and stored at −80 °C for DNA extraction. In addition, 13 fresh leaf samples were

collected from grazing pastures (2 samples), sugarcane (6 samples), forest (3 samples) and banana (2 samples) of the Johnstone catchment following standard procedures[53]. Eight litter samples were also collected from the ground of sugarcane (4 samples), forest (2 samples) and banana (2 samples) land uses within the Johnstone catchment. Leaf and litter samples were stored in paper bags, protected from heat, and placed in a refrigerator before being transported to the laboratory. Subsamples of collected leaf and litter samples were oven-dried for 5 days at 65 °C and finely ground (<150 μm) prior to laboratory analysis. In the Burdekin catchment, nine surface soil samples (0–10 cm) were collected from grazing pasture (6 samples) and forest (3 samples) land uses. Four sub-soil samples (10–20 cm) were also collected from the grazing pastures of the Burdekin catchment.

### Riverine and estuarine SPM collection

The SPM were collected from freshwater sections (riverine), inner estuarine mixing zone (IEMZ) and outer estuarine mixing zone (OEMZ) of the Johnstone, Tully and Burdekin Rivers. In general, the separation of these estuarine mixing zones are related to changes in SPM concentration and salinity and are broadly related to primary and secondary colour class water types established in remote sensing studies[37,54]. A total of nine riverine (Johnstone and Tully: 2 samples, Burdekin: 7 samples), seven IEMZ (Johnstone and Tully: 2 samples, Burdekin: 5 samples) and 11 OEMZ (Tully: 3 samples, Burdekin: 8 samples) SPM samples were collected. All the SPM samples (riverine, IEMZ and OEMZ) were collected during the flood plume events during the wet seasons (January to March) of 2017, 2018 and 2019 to capture pulsed inputs of SPM during flood events with the potential to reach out well beyond the inner GBR and into the mid-shelf area (Fig. 1). The SediPump® system[55] was used to collect required SPM amounts across the transition from the freshwater to estuarine and marine environments[37]. This sampling system allows large volumes of water (9,000–12,000 L) to be pumped through a 1 μm filter cartridge (Puretec® sediment filter cartridge-wound GW011) over a relatively short period (2–3 h). The sediment retained in the filter carousel was recovered by cutting the string filter cartridge to release the sediment (Fig. S1). Homogenous subsamples from the riverine, IEMZ (Johnstone River: 2 samples, Tully River: 1 sample, Burdekin River: 3 samples) and OEMZ (Tully River: 1 sample, Burdekin River: 2 samples) zones were stored at -80 °C for DNA extraction.

### Marine trap sediment collection

The marine traps[56] were installed 50 cm off the seafloor (Fig. S2) along the main river catchments of the GBR (Fig. 1). The GBR stretches over a significant area including inshore, mid-shelf and offshore regions, however, in this study the marine traps were installed within the inner GBR shelf to capture sediment resuspension events (dry season deployment) as well as the influence of SPM delivered through the riverine flood plumes (during the wet season deployments). An array of three to four marine traps was established at seven sites off the coast from the Burdekin catchment in Cleveland Bay (e.g., Middle Reef, Cleveland Bay, Orchard Rocks, Geoffrey Bay) and Halifax Bay (e.g., Orpheus, Pelorus and Havannah Islands) to capture the SPM exposure in the Dry Tropics region including the influence of the Burdekin River flood plume (Fig. 1). Similarly, the Dunk Island marine trap site captured the SPM originating from the Wet Tropics region including the flood plume from the Tully River. The traps were deployed continuously at these sites over a three (Dunk Island) to four (Cleveland-Halifax Bay) year period and the traps were changed-over every 3–4 months to capture seasonal changes in SPM composition on the inner GBR. A total of 82 SPM samples captured by marine traps (Dunk Island site: 12 samples; Cleveland and Halifax Bays sites: 70 samples) were analysed for organic composition between 2016 and 2019. Of these, 16 samples were collected during the wet season and influenced by riverine flood plumes (influenced trap sediment, ITS), while

66 samples were collected during dry season intervals and were not influenced by riverine flood plumes (not influenced trap samples, NTS). Nineteen homogenous subsamples of NTS (Dunk Island: 2 samples, Halifax Bay: 4 samples, Cleveland Bay: 13 samples) were frozen and stored at −80 °C for DNA extraction.

The term "SPM" represents the sediment exclusively transported via rivers including within the rivers themselves and within riverine flood plumes. The term "marine sediment" represents the sediment captured within sediment traps that were deployed at various times coinciding with or without the influence of flood plumes. Hence any sediment captured by the marine trap during the deployment period represent sediment that has at some stage been suspended in the water column through either resuspension or settling in the flood plumes.

### Analysis of organic functional groups

Solid-state $^{13}$C nuclear magnetic resonance ($^{13}$C NMR) spectroscopic analysis was used to identify the OC functional groups of plant material, soil and SPM across the catchment to reef continuum. Soil and SPM samples were pre-treated with hydrofluoric acid (HF) prior to solid-state $^{13}$C NMR spectroscopic analysis to remove paramagnetic species (e.g., $Fe^{3+}$ and $Mn^{2+}$), minimise their impacts on NMR spectra and concentrate the OM of the whole soil and SPM. For this pre-treatment stage, 5 g of each sample was placed into a Nalgene high speed centrifuge tube, then 30 ml of 10% HF was added to each tube. The tubes were shaken on an end-to-end shaker (2 h), and then centrifuged (10,000 rpm or -17,500 × $g$, 15 min). Following centrifugation, the supernatant was separated from the solid residue, neutralised to pH 6–8 with $CaCO_3$ powder and then discarded. The HF treatment procedure was repeated a further four times for each sample, however, for the final extraction the mixture was shaken for 16 h (overnight). Following the final HF extraction, the shaking/centrifugation procedure was repeated six times for each of the treated samples using distilled water, to ensure the remaining residual HF was removed. The resulting treated/washed samples were then dried (60 °C, 48 h) prior to $^{13}$C NMR spectroscopic analysis.

The $^{13}$C NMR spectra of leaf samples were acquired using a 400 MHz Varian INOVA spectrometer (Varian Inc., CA) operating at a $^{13}$C frequency of 100.6 MHz. Samples were packed in a 7 mm diameter silicon nitride rotor and spun at 5 kHz at the magic angle. The cross-polarisation sequence xpolar1, contained within the VnmrJ 2.1B software package was used. A total of 2,000 transients were collected for leaf samples. A contact time of 2 ms, an acquisition time of 14 ms, a recycle delay of 2.5 s and a spectral width of 500 ppm were used in all cases. Spectra were processed using MestReNova v11.0 and lorentzian line broadening functions of 20 Hz were applied to all spectra. The $^{13}$C NMR spectra of plant litter, soil and SPM were acquired using a 300 MHz Varian VNMRS spectrometer (Varian Inc., CA) operating at a $^{13}$C frequency of 75.4 MHz. Samples were packed in a 7 mm diameter silicon nitride rotor and spun at 5 kHz at the magic angle. The Tangent Cross Polarisation (tancpx) sequence within the VnmrJ 3.1 A software package was used to perform the cross polarisation which was a successor of the xpolar1 sequence in the older version (VnmrJ 2.1B). A total of 20,000 transients for soil and SPM samples and 2,000 transients for plant litter samples were collected. A contact time of 1.2 ms, an acquisition time of 20 ms, a recycle delay of 2.5 s and a spectral width of 477 ppm were used in all cases. Spectra were processed using the MestReNova v11.0 software package (Mestrelab Research S.L.). Lorentzian line broadening functions of 50 Hz were applied to all spectra. $^{13}$C chemical shift values were referenced relative to external hexamethylbenzene (HMB; $\delta_{CH3}$, 17.4 ppm).

To quantify different OC functional groups, the following chemical shift boundaries were selected for the $^{13}$C NMR spectra based on Prietzel et al.[57]. The alkyl C functional group (0–45 ppm) represented aliphatic and recalcitrant organic compounds such as lipids, cutin,

suberin and amino acids. The O-alkyl C was within the chemical shift region 45–110 ppm representing labile organic compounds such as carbohydrates, cellulose and hemicellulose. The chemical shift boundary 110–160 ppm included the aryl C representing aromatic compounds such as lignin and tannin, while the chemical shift region 160–180 ppm represented the carboxyl functional group. The alkyl C (0–45 ppm) to O-alkyl C (45–110 ppm) ratio (A/O-A ratio) of the $^{13}C$ NMR spectra provides an indicator of the decomposition process[58]. Representative solid-state $^{13}C$ NMR spectra for plant litter, soil and SPM samples are provided in Fig S3. In this study, the chemical composition of OM was investigated in all collected soil, leaf and litter samples. The $^{13}C$ NMR spectra were acquired for a total of 15 SPM samples including six riverine (Johnstone and Tully Rivers: 2 samples, Burdekin River: 4 samples), 5 IEMZ (Tully and Johnstone Rivers: 2 samples, Burdekin River: 3 samples) and 4 OEMZ (Tully River: 2 samples and Burdekin River: 2 samples). A total of 42 marine trap sediment samples were also analysed for $^{13}C$ NMR, of which 10 samples were collected during the wet season and influenced by flood plumes (influenced trap sediment, ITS), while 32 samples were collected within dry season intervals and not influenced by flood plumes (not influenced trap sediment, NTS).

## Stable isotope and elemental analysis

For the stable isotope nitrogen analysis ($\delta^{15}N$), all soil and SPM samples were pelletized in tin capsules. For stable isotope carbon ($\delta^{13}C$) analysis, inorganic C was first removed by 10% hydrochloric acid (HCl), then the samples were pelletized in silver capsules and weighed prior to analysis by a Sercon Hydra 20–22 Europa EA-GSL isotope-ratio mass spectrometer. Stable isotope ratios are reported in standard delta ($\delta$) notation per mil (‰) as: $\delta_X = [R_{sample}/R_{standard} - 1] \times 1000$ where $X$ is $^{13}C$ or $^{15}N$ and $R = {}^{13}C/{}^{12}C$ or $^{15}N/^{14}N$, respectively. Standard reference materials were PDB limestone for C and Air was the standard for N[59]. Plant materials were also analysed for $\delta^{13}C$ and $\delta^{15}N$ without any pre-treatment. Chemical elements (e.g., Na, K, Ca, Mg, Cr, Ni, Mn, Co, Al, Fe and P) were analysed in soils and SPM, using ICP-OES (Perkin Elmer; Optima 8300) after direct digestion with nitric and perchloric acid following Miller[60].

## DNA extraction and high-throughput sequencing

The total genomic DNA was extracted from 0.3–0.5 g of frozen soil and SPM using MoBio Powersoil DNA isolation kit following the manufacturer's instructions. A NanoDrop ND-2000 UV–vis spectrophotometer (NanoDrop Technologies, Wilmington, DE, USA) and agarose gel electrophoresis were used to assess the concentration and quality of extracted DNA, respectively. Primers 515F/806R and ITS1F/2043R[61] targeting the V4 region of bacterial 16S rRNA gene and fungal Internal Transcribed Spacer (ITS) region were used for Polymerase Chain Reaction (PCR) analysis, respectively. PCR was conducted in a total volume of 30 μL containing 15 μL of Phusion® High-Fidelity PCR Master Mix (NEB, Ipswich, MA, USA), 0.2 mM of forward and reverse primers, and ~10 ng DNA template; each PCR run included a negative control. Purified PCR products and the sequence library were prepared according to Du et al.[62], and the sequencing was performed on an Illumina Miseq sequencing platform (Illumina, San Diego, CA, USA). The obtained raw sequences were quality filtered, assembled, de-multiplexed, and assigned to individual samples using Quantitative Insights into Microbial Ecology (QIIME) analysis[63]. A chimera filtering approach (UPARSE) was used to bin the sequences into operational taxonomic units (OTUs) at the 97% sequence identity level[64]. Approximately 3.8 M and 3.5 high-quality merged sequences were mapped for 16S rRNA and ITS OTUs, respectively. Representative sequences of 16S rRNA and ITS OTUs were annotated against the Silva[65] and UNITE[66] databases in QIIME using the UCLUST algorithm[67], respectively. Ambiguous taxonomic predictions (unassigned) were further checked against the Reference Sequence Database in GenBank. To minimise the variation caused by the sampling sequences, each

sample was rarefied by 23,100 and 16,000 sequences for bacteria and fungi, respectively. All 16S rRNA gene and ITS fragment sequences were deposited in the Sequence Read Archive (SRA) with the accession number PRJNA690602.

## Bacteria to fungi gene copy number ratio

The abundances of bacterial 16S rRNA genes and fungal ITS copy numbers were measured in SPM collected from riverine, IEMZ, OEMZ and marine trap sediment by quantitative Polymerase Chain Reaction (qPCR) using SYBR1 Premix Ex Taq™ II (Takara, Japan) on a LightCycler1 96 Real-time PCR System (Roche, Switzerland). The qPCR reaction (20 μl) contained 10 μl of SYBR Premix Ex Taq™ II, 0.4 μl each of the forward and reverse primers, 2 μl of template DNA and 7.2 μl of sterile ultrapure water. The PCR cycling conditions were as follows: 45 cycles of 95 °C for 10 s, primer annealing temperature for 30 s and template extension at 72 °C for 45 s. The PCR products from each gene were purified with a Gel Extraction Kit (CW Biotech Co, Beijing, China) and cloned into pMD19-T vector (Takara, Japan). Plasmids from the positive clones with the target gene insert were extracted with a Pure-Plasmid Mini Kit (CW Biotech Co., Beijing, China). The concentration of plasmid was determined on a NanoDrop ND-2000 UV–Vis spectrophotometer (NanoDrop Technologies, Wilmington, DE, USA), and used for the calculation of standard copy numbers. Ten-fold serial dilutions of plasmid in triplicate were used to generate a standard curve for each gene and to check the amplification efficiency. For each gene, a high amplification efficiency of 92–98% was obtained, the $R^2$ values were >0.992 and no signal was observed in the negative controls.

## Statistical analysis

Structural equation modelling (SEM) was conducted using the "lavaan" R package version 0.6-10[68] to assess the complex inter-relationships among microbial diversity and community structure (MDCS), environmental factors (Env1 and Env2), organic matter quality (OMQ) and organic carbon characteristics (OCC) in soil, riverine, IEMZ, OEMZ and marine sediment traps. The OMQ was included in the SEM analysis as a latent factor comprising a range of OM characteristics (TOC, TOC:TN ratio, $\delta^{13}C$, $\delta^{15}N$, carboxyl, aryl, O-alkyl and alkyl C) that reflected the quality of OM in soil, riverine, estuarine and marine habitats. In addition, the proportion of O-alkyl, alkyl and aryl C functional groups were included in the SEM analysis as a separate latent factor of key OCC in terrestrial and marine environments. The MDCS was included in the SEM as a latent factor of biological parameters including bacterial and fungal richness (BactRich and FunRich), diversity (Shannon Index - BactShann and FunShann) and community structure (BactNMDS and FunNMDS) of soil, riverine, IEMZ, OEMZ and marine trap sediment. All the interactive pathways among latent factors were considered in the initial SEM model. A modified model was constructed by removing non-significant and less influential pathways when the hypothesised full model did not produce an adequate fit.

The network analysis was performed using the SparCC algorithm which is capable of estimating correlation values from a rich ecological network connecting interacting species[69,70]. To minimise the influence of rare taxa, only OTUs with more than four observations were kept. The False Discovery Rate (FDR) was controlled by performing 1,000 bootstraps. Only the strong ($p > 0.60$) and robust ($p < 0.001$) correlations were retained. Nodes were coloured according to the modules. The networks were displayed in the Gephi software[71]. Network analysis generated 24,909 significant correlations ($p < 0.01$) between microbial OTUS and soil and SPM characteristics (as explained above). Metabolic pathway abundance prediction, based on the METACYC database, was performed using PICRUST2[72] applying the default parameters. In the final network, 1,039 edges were kept in the network after increasing the robustness (RHO) from 0.2 to 0.6. The non-metric

multidimensional scaling (NMDS) ordination was conducted based on Bray-Curtis distance with the relative abundance of OTUs using "Vegan" package in R version3.5.3[73]. Spearman's rank correlations among different factors were constructed to visualise the correlations, $P < 0.05$ as statistically significant.

## Reporting summary

Further information on research design is available in the Nature Portfolio Reporting Summary linked to this article.

## Data availability

All 16S rRNA gene and ITS fragment sequences were deposited in the Sequence Read Archive (SRA: https://www.ncbi.nlm.nih.gov/sra/PRJNA690602) with the accession number "PRJNA690602. The data for bioavailability and chemical composition of the organic component of soils and SPM from the catchment to reef project (NESP TWQ 5.8) has been submitted to e-Atlas data catalogue (Australian Institute of Marine Science) and can be accessed at https://eatlas.org.au/data/uuid/3dadf670-f9b1-44d8-bce0-a5aeddf2c20a.

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

## Acknowledgements

This work was financially supported by the Australian Government's National Environmental Science Program (NESP projects 2.1.5 and 5.8 to S.L. and Z.B.), Griffith University (M.B. and C.R.C.) and the Great Barrier Reef Marine Park Authority Reef Guardian Research Grant 2019 (M.B and C.R.C.). We acknowledge Joanne Burton and Alex Garzon-Garcia (Department of Environment and Science) for supplying the Burdekin soil samples and Sue Boyd and Horst Joachim Schirra for their help in using Magnetic Resonance Facilities at Griffith University. We thank Dr. Yan Gao from the Institute of Agricultural Resources and Environment, Jiangsu Academy of Agricultural Sciences, Nanjng, China for analysing soil DNA sequences. The field sediment traps, and instrument loggers were deployed under permits G17/38148.1 and G16/38774.1 issued by the GBRMPA. We thank Simon Griffiths, Andreas Dietzel, Adam Wilkinson, Cassandra Thompson, Sofia Valero Fortunato, Blanche Danastas, Ian McLeod, Lauren Firby, Eridani Mulder, Jane Waterhouse, Sandra Erdmann, Kai Pacey and Glen Ewels for being part of the dive team who collected the marine trap samples. The Marine Geophysics Laboratory is thanked for supplying and helping calibrate the nephelometers.

## Author contributions

M.B. and C.R.C. conceived the original idea, developed conceptual framework and designed the paper. M.B. performed laboratory analysis, statistical analysis and wrote the original draft. M.B., C.R.C., S.L., M.R.R., Z.B. and T.S. performed field sampling. J.W., J.S. and M.B. performed bioinformatic analysis. J.W. did the network analysis. E.H. did the SEM model. Q.H. performed DNA sequencing analysis. All of the authors contributed to the manuscript revision.

## Competing interests

The authors declare no competing interests.
