## [Peer review file · Nature Communications]

REVIEWER COMMENTS

Reviewer #1 (Remarks to the Author):

This article shows that the suspended particulate matter in the Great Barrier Reef differ significantly from the suspended particulate matter of terrestrial origin and is produced by marine phytoplankton, whereas the terrestrially-derived suspended particulate matter is decomposed in the riverine and estuarine mixing zones before it is transported offshore. This conclusion was reached by combining various methods, including the determination of isotopic and genetic fingerprints as well as the analysis of organic functional groups of a wide range of samples, including soils, terrestrial plant litter, suspended particulate matter collected from riverine as well as inner and outer estuarine mixing zones and marine sediments.

The manuscript may be publishable in Nature Communications provided the authors address the points below.

p. 8, l. 182: the authors must define the microbial diversity and community structure and cite a reference providing more details on this concept.

p. 13, l. 299: the authors must explain why they did not collect suspended particulate matter in the marine environment but only marine sediment. They must discuss in the article the possible differences in chemical composition, isotopic and genetic fingerprints between marine sediments and marine suspended particulate matter.

p. 15, ll. 338 and 347: the authors must explain what the tancpx and xpolar1 sequences are, how they differ and why they employed different pulse sequences to record ¹³C NMR spectra of plant litter, soil and suspended particulate matter on the one hand, and leaf samples on the other hand.

p. 15, ll. 341 and 350 : the authors must indicate the employed radiofrequency fields and they must clarify which parameter is swept over the given sweep widths.

p. 18, l. 430 : the authors must cite a reference for the « lavaan » R package.

Figure 2: ellipsoids must be defined.

Figure 2: the figure does not display red points.

Figures 3 and 4c, d and e: the meaning of letters a, b and c must be clarified.

Figure 5: The acronyms DON and DOC must be expanded.

Table S1 : are the superscripted letters labels of the footnotes?

Figures S6a and S7b: the authors must explain what are the different vertical bars. Do they correspond to different samples?

Table S3: are the superscripted letters labels of the footnotes?

A few typos:

Caption of Figure 2: UEMZ

SI, l. 101: a typo: Fig. 8

Reviewer #2 (Remarks to the Author):

The manuscript “The origin of suspended particulate matter in the Great Barrier Reef” is well-written and represents a significant amount of work. The authors hypothesise that the suspended particulate matter (SPM) on the GBR does not have a terrestrial but marine origin. Although the authors’ hypothesis may be correct, I think that their study falls short of providing compelling evidence to reassign the origin of the SPM on the GBR. Given the significant implication of their conclusion in terms of water quality management, the data needs to be rock solid.

The main issue of this study is that the marine origin of the SPM is only based on the fact that the SPM composition of the marine samples is dissimilar to the terrestrial sources. But this approach misses half of the picture. Indeed, ascribing the source would require concomitantly showing that the SPM becomes more similar to offshore (outer reefs; not affected by terrestrial input). Instead, the authors solely rely on previous measurements of phytoplankton-derived organic matters from Meyers and Limoges et al. (note: this last paper does not appear in the reference list of the manuscript) to ascribe the phytoplankton origin of the SPM on the GBR.

Unlike the productive waters of Rhode Island and Washington that were used by Meyers for his measurements, the GBR water is much more oligotrophic, with phytoplankton communities dominated by cyanobacteria (Furnas and Mitchell 1986). The contribution of phytoplankton-derived organic carbon to the total dissolved organic carbon within coral reef environments is typically low (Alldredge et al. 2013; Cardini et al. 2016).

In addition, no significant difference is apparent in the different parameters measured between the OEMZ samples and the sediment trap samples (influenced or not by flood plume). The only differences are between the microbial profiles. This means that the marine samples do not seem to be dissimilar, even though some of them are more influenced by riverine input than others (i.e., ITS vs NTS). I find this very puzzling.

Finally, some of the conclusions derived from the structural equation modelling are misleading. This analysis relies on correlations between your variables, and causation should not be implied when discussing the results. It is therefore incorrect to state that “the microbial diversity and community structure (MDCS) mediated the effects of habitat types (line 182)”, or “The shift in MDCS was due to the significant variation in mineral component (line 184)”. Lastly, using 16S and 18S amplicon sequencing, it is not correct to conclude anything regarding the efficiency and biological capacity of the microbial communities, such as line 202: “transportation pathway (soil-riverine-estuarine-marine) moves towards a microbial community structure with higher efficiency and biological capacity to survive on complex and stable forms of OM”. If the authors want to discuss the metabolic capacity of the microbial communities, their full genomic makeup (metagenomes) needs to be analysed.

Reviewer #3 (Remarks to the Author):

This paper builds on suspended sediment and soil data collected at numerous sites in northern Queensland (AU) across catchments, river channels and marine environments (including inshore and mid-shelf GBR regions). The data indicate “the fingerprints of SPM in the marine environment were completely different from those of terrestrial origin but more consistent with that formed by marine phytoplankton”. From these observations the hypothesis is derived that “the SPM in the GBR may not have terrestrial origin but produced locally in the marine environment”. This hypothesis, in turn, leads authors of this paper to the conclusion there are needs for “a new conceptual framework that adequately recognises and addresses both terrestrial and marine sources of SPM in the GBR”. This new framework is expected to underpin a refined Water Quality Improvement Plan (WQIP) on GBR.

I believe, the readership of this paper would benefit from clarifying several questions relevant to this study. For example, the discussion in this paper is focused on the carbonate fraction of the SMP. Very little is said about the mineral component of the SPM. It would help to elucidate the role of the mineral particles in GBR. Does it matter how much such mineral particles are discharged from catchments?

How to translate findings in this paper to other regions of GBR particularly those with a high energy tidal environment?

Another point to clarify is the dynamic nature of sediment processes on GBR particularly during the episodic flood events which can generate flood plums reaching out well beyond the inner shelf zone and into the mid-shelf regions (consistent with the remote sensing data). The data reported in this paper (and sampled over the course of several years) represent an average condition on the GBR. Hence, pulsed inputs of SPM during flood events operating over the time scales from hours to days may not

necessarily fit the conceptual picture of the sediment transformation drawn in this study from such data. I believe, the readership of this paper would benefit from a further discussion of this point.

GBR is covering a huge area including inshore, mid-shelf and offshore regions. For a paper discussing impacts of catchments on GBR it would help to clarify which regions of the GBR are being considered.

I think, the hypothesis that “the SPM in the GBR may not have terrestrial origin but produced locally in the marine environment” is justified by the data presented in this paper and could be valid for a typical day on GBR. On the other hand, during flood events, this hypothesis may not hold (at least, no data were presented in this paper to support it). Analogously, the conceptual model presented in this paper and describing the transformation of the SPM across the freshwater-marine environments may not be valid during such extreme weather events. I think, this paper will be improved by acknowledging such a limited scope of this hypothesis.

To conclude, I think data presented in this paper warrant further research into SPM transformations across the freshwater and marine environments of GBR particularly during the high flood events. The outcomes of such study may lead to a new conceptual model to underpin a refined WQIP.

Observations reported in this study highlight specific pathways of the SPM transformation across the freshwater and marine environments in GBR stimulating new hypothesis about sediment transformation in such systems. This study will be of interest to the practitioner working in GBR and other reef systems elsewhere. I think, after a minor revision this study can be published in the journal.

Typos:

Fig.2 captions – replace UEMZ with OEMZ

References 2 and 3 have incomplete list of co-authors.

RESPONSE TO REVIEWERS' COMMENTS

Reviewer #1 (Remarks to the Author):

Question (Q): p. 8, l. 182: the authors must define the microbial diversity and community structure and cite a reference providing more details on this concept.

Response (R): The microbial diversity and community structure was defined and references were added (pages 8, lines: 182-185, references 33 and 34):

“The biological component of SPM represented by microbial diversity and community structure (MDCS), consisting of bacterial and fungal richness (BactRich and FunRich), diversity (Shannon Index - BactShann and FunShann) and community structure (BactNMDS and FunNMDS) (33, 34)”.

(Q): p. 13, l. 299: the authors must explain why they did not collect suspended particulate matter in the marine environment but only marine sediment. They must discuss in the article the possible differences in chemical composition, isotopic and genetic fingerprints between marine sediments and marine suspended particulate matter.

(R): We realise there is some confusion in the terminology used in our manuscript. We used the term “suspended particulate matter” in the text to represent the sediment exclusively transported via rivers including within the rivers themselves and within riverine flood plumes. The term “marine sediment” was used to define the sediment captured within sediment traps that were deployed at various times coinciding with or without the influence of flood plumes. Indeed, the sediment traps were deployed ~ 50 cm off the bottom of the seafloor and hence any sediment captured by the trap during the deployment period represent sediment that has at some stage been suspended in the water column through either resuspension (*i.e.*, remobilised from the seafloor predominantly from waves) or settling in the flood plumes. Hence this marine sediment also technically represents suspended particulate matter.

We have added a paragraph to the manuscript to more clearly define these different collection methods (page 15, lines 336-341):

“The term “SPM” represents the sediment exclusively transported via rivers including within the rivers themselves and within riverine flood plumes. The term “marine sediment” represents the sediment captured within sediment traps that were deployed at various times coinciding with or without the influence of flood plumes. Hence any sediment captured by the marine trap during the deployment period represent sediment that has at some stage been suspended in the water column through either resuspension or settling in the flood plumes”.

(Q): p. 15, ll. 338 and 347: the authors must explain what the tanctx and xpolar1 sequences are, how they differ and why they employed different pulse sequences to record ¹³C NMR spectra of plant litter, soil and suspended particulate matter on the one hand, and leaf samples on the other hand.

(R): The tanctx sequence in the VnmrJ 3.1A software is the successor/replacement of the xpolar1 sequence in the older Vnmr 2.1B software. They perform the same cross-polarisation experiment.

We have clarified this point further in the method section (page 16, lines 369-371):

“The Tangent Cross Polarization (tanctx) sequence within the VnmrJ 3.1A software package was used to perform the cross polarisation which was a successor of the xpolar1 sequence in the older version (VnmrJ 2.1B)”.

(Q): p. 15, ll. 341 and 350: Which parameter is swept over the given sweep widths.

(R): The “sweep width” is another name for the “spectral width”, i.e., for the width of the recorded spectrum. To avoid confusion the term “sweep width” has now been replaced with “spectral width”, and the spectral widths are stated in ppm (page 16, lines 364 and 370).

(Q): p. 18, l. 430: the authors must cite a reference for the « lavaan » R package.

(R): Reference was added (page 19, line 456, reference 68).

(Q): Figure 2: ellipsoids must be defined and figure does not display red points.

(R): Ellipses were defined and red points were added to Fig. 2.

(Q): Figures 3 and 4c, d and e: the meaning of letters a, b and c must be clarified.

(R): The meaning of letters was added to both figures.

“Box plots with the same letter (*e.g.*, a, b, c) are not significantly different at $p < 0.05$ ”.

(Q): Figure 5: The acronyms DON and DOC must be expanded.

(R): Dissolved organic carbon (DOC) and dissolved organic nitrogen (DON) were added to Fig. 5 caption.

(Q): Table S1: are the superscripted letters labels of the footnotes?

(R): Letters a, b and c are to show significance if difference at $p < 0.05$. Table’s footnote was rewritten to clarify the letters.

“Means (SD) within a column followed by the same letter are not significantly different at $p < 0.05$ ”.

(Q): Figures S6a and S7b: the authors must explain what are the different vertical bars. Do they correspond to different samples?

(R): The explanation for bars was added Figs. S6 and S7 captions.

“Each bar represents a sample”.

(Q): Table S3: are the superscripted letters labels of the footnotes?

(R): Letters a, b and c are to show significance if difference at $p < 0.05$. The table’s footnote was rewritten to clarify the letters.

“Means (SD) within a column followed by the same letter are not significantly different at $p < 0.05$ ”.

(Q): Caption of Figure 2: UEMZ

(R): UEMZ was corrected OEMZ.

(Q): SI, l. 101: a typo: Fig. 8

(R): Fig. 8 was changed to Fig. S8

Reviewer #2 (Remarks to the Author):

(Q): The manuscript “The origin of suspended particulate matter in the Great Barrier Reef” is well-written and represents a significant amount of work. The authors hypothesise that the suspended particulate matter (SPM) on the GBR does not have a terrestrial but marine origin. Although the authors’ hypothesis may be correct, I think that their study falls short of providing compelling evidence to reassign the origin of the SPM on the GBR. Given the significant implication of their conclusion in terms of water quality management, the data needs to be rock solid.

(R): This study, for the first time, provides three advanced fingerprinting evidence (isotopic, ^{13}C -NMR spectroscopy and genetic fingerprints) across multiple samples collected along the terrestrial-estuarine-marine continuum of the GBR. All tested fingerprints have strongly indicated a significant change in microbial communities as well as isotopic (^{13}C) and chemical structure of organic matter into a complex form which is completely different from their terrestrial origin and similar to that of produced by marine phytoplankton. We concede that sediment deposited near the river mouths (within the inner estuarine mixing zone) of the flood plume has not been sampled during and after flood periods to examine how the terrestrial organic matter is degraded over time (we recommend this be subject to future study). While studies of the processing of terrestrial organic matter in river estuaries of the GBR (e.g., Radke et al., 2010; Crosswell et al., 2020) have documented the rapid microbial degradation in such locations so far a detailed fingerprinting analysis of these sediments have not been performed. In any case, our data show that there is a rapid change in the chemical makeup of the suspended particulate matter between the inner estuarine mixing zone and the outer estuarine mixing zone, which highlight that terrestrial organic matter inputs are almost exclusively deposited near river

mouths with very little (if any) even reaching our inshore monitoring sites. This confirms previous findings (e.g. Gagan et al., 1987) which simply measured for the presence of organic matter but did not undertake the comprehensive fingerprinting analysis performed in our study.

We have modified the manuscript to incorporate this conceptual understanding (page 10, lines 218-227):

“In this study sediment deposited near the river mouths (within the IEMZ) of the flood plume has not been sampled during and after flood periods to examine how the terrestrial OM is degraded over time. However, previous studies on the processing of terrestrial OM in river estuaries of the GBR (41, 42) have documented the rapid microbial degradation in such locations. Our detailed fingerprinting analysis, for the first time, show that there is a rapid change in the isotopic, structural and genetic fingerprints of the SPM between the IEMZ and the OEMZ, which highlight that terrestrial organic matter inputs are almost exclusively deposited near river mouths with very little (if any) even reaching our inshore monitoring sites”.

(Q): The main issue of this study is that the marine origin of the SPM is only based on the fact that the SPM composition of the marine samples is dissimilar to the terrestrial sources. But this approach misses half of the picture. Indeed, ascribing the source would require concomitantly showing that the SPM becomes more similar to offshore (outer reefs; not affected by terrestrial input). Instead, the authors solely rely on previous measurements of phytoplankton-derived organic matters from Meyers and Limoges et al. (note: this last paper does not appear in the reference list of the manuscript) to ascribe the phytoplankton origin of the SPM on the GBR.

(R): Our results showed that the ^{13}C and TOC:TN ratio of SPM collected from the OEMZ were similar to the SPM collected offshore in the marine environment (NTS; not affected by terrestrial inputs) which confirms the marine origin of SPM in the GBR (Fig. 2).

We added below paragraph to further highlight and clarify this point (page 5, line 105-107):

“Moreover, the $\delta^{13}\text{C}$ and TOC:TN ratios of SPM collected from OEMZ fell within the ranges measured in the marine trap samples not influenced by flood plumes (Fig. 2)”.

(R): Reference (A. Limoges . A et al) was added to the manuscript (page 5, line 105, reference 22).

(Q): In addition, no significant difference is apparent in the different parameters measured between the OEMZ samples and the sediment trap samples (influenced or not by flood plume). The only differences are between the microbial profiles. This means that the marine samples do not seem to be dissimilar, even though some of them are more influenced by riverine input than others (i.e., ITS vs NTS). I find this very puzzling.

(R): The similarity between fingerprints of SPM in OEMZ and ITS and NTS further confirms the marine origin of the SPM in the GBR. This result indicates that the phytoplankton biomass and marine primary production is significant at OEMZ and marine environment and significantly contribute to the SPM in the GBR.

Please see above response. We added a paragraph to further highlight and clarify this point (page 5, line 105-107).

(Q): Finally, some of the conclusions derived from the structural equation modelling are misleading. This analysis relies on correlations between your variables, and causation should not be implied when discussing the results. It is therefore incorrect to state that “the microbial diversity and community structure (MDCS) mediated the effects of habitat types (line 182)”, or “The shift in MDCS was due to the significant variation in mineral component (line 184)”.

(R): Line 182 was rewritten and corrected accordingly (page 8 lines 185-187):

“Results showed that the shift in MDCS was highly correlated with the variation in environmental factors (Env2) and organic carbon characteristics (OCC) of the SPM across habitat types (Fig. 4a).”

Line 184: this paragraph was rewritten and corrected accordingly (page 8 lines 187-193).

“Network analysis indicated that the shift in microbial community composition across habitats was particularly influenced by the variation in environmental factors such as the characteristics of mineral component (e.g., particle size distribution) and solution chemistry (e.g., salinity, pH) of the mediums in which SPM was transported along the soil-riverine-estuarine-marine continuum (Fig. 4b).”

Lastly, using 16S and 18S amplicon sequencing, it is not correct to conclude anything regarding the efficiency and biological capacity of the microbial communities, such as line 202: “transportation pathway (soil-riverine-estuarine-marine) moves towards a microbial community structure with higher efficiency and biological capacity to survive on complex and stable forms of OM”. If the authors want to discuss the metabolic capacity of the microbial communities, their full genomic makeup (metagenomes) needs to be analysed.

This line was rewritten to better represent the results (page 9, lines 205-207):

“Thus, such a transportation pathway (soil-riverine-estuarine-marine) moves towards a microbial community structure in the marine environment which is significantly different from terrestrial origin (Fig. 4d)”.

Reviewer #3 (Remarks to the Author):

(Q): I believe, the readership of this paper would benefit from clarifying several questions relevant to this study. For example, the discussion in this paper is focused on the carbonate fraction of the SMP. Very little is said about the mineral component of the SPM. It would help to elucidate the role of the mineral particles in GBR. Does it matter how much such mineral particles are discharged from catchments?

(R): In this study we analysed the bulk component of the SPM collected in flood plumes and sediment traps which represent a combination of mineral, carbonate and organic matter components (Bainbridge et al 2021).

However, to further clarify this point, the role of terrestrially derived sediment and relevant literature was added to discussion (page 11, lines 238-240).

“Several studies have examined and identified the role that terrestrially-derived SPM plays in affecting rates of sedimentation and water clarity in the riverine and inner estuarine environments (3, 37, 45), however, further research is urgently required to identify the key environmental factors driving the production of organic-rich SPM in the outer estuarine and marine environments”.

(Q): How to translate findings in this paper to other regions of GBR particularly those with a high energy tidal environment?

(R): The result from this study is extendable to all river estuaries discharging into the marine environment including high tidally driven estuarine zones. There have been some studies that have examined the degradation/processing of organic matter and associated particulate nutrients in high tidally driven estuarine zones which also show rapid cycling of nutrients (e.g. Radke et al., 2010; Crosswell et al., 2020). The Radke et al. (2010) in particular shows that the highest rate of organic matter degradation/cycling occurs in the zone of maximum resuspension within the estuary.

We have added an additional sentence and references in the manuscript to capture this point (page 10, lines 221-223, references 41, 42):

“Previous studies of the processing of terrestrial OM in river estuaries of the GBR have documented the rapid microbial degradation in such locations including high tidally driven estuarine zones (41, 42)”.

(Q): Another point to clarify is the dynamic nature of sediment processes on GBR particularly during the episodic flood events which can generate flood plumes reaching out well beyond the inner shelf zone and into the mid-shelf regions (consistent with the remote sensing data). The data reported in this paper (and sampled over the course of several years) represent an average condition on the GBR. Hence, pulsed inputs of SPM during flood events operating over the time scales from hours to days may not necessarily fit the conceptual picture of the sediment transformation drawn in this study from such data. I believe, the readership of this paper would benefit from a further discussion of this point.

(R): We note that in this study all the SPM samples were collected during the flood plume events within the wet season to capture pulsed inputs of SPM during flood events which included sampling on the mid shelf (Old Reef) during a flood plume from a very large flood plume derived from the Burdekin River. The temporal coverage of the sediment trap deployments also covers intervals that coincide with large flood events as well as during the dry season which lack any considerable river inputs. Hence our study has not only captured the ‘average condition’ but also the large flood events.

A paragraph was added to further clarify this point (page 13, lines 301-304):

“All the SPM samples (riverine, IEMZ and OEMZ) were collected during the flood plume events during the wet seasons (January to March) of 2017, 2018 and 2019 to capture pulsed inputs of SPM during flood events with the potential to reach out well beyond the inner GBR and into the mid-shelf area (Fig. 1).”

(Q): GBR is covering a huge area including inshore, mid-shelf and offshore regions. For a paper discussing impacts of catchments on GBR it would help to clarify which regions of the GBR are being considered.

(R): A paragraph was added to further clarify this point (page 14, lines 315-320).

“The marine traps were installed 50 cm off the seafloor (Fig. S2) along the main river catchments of the GBR (Fig. 1). The GBR stretches over a significant area including inshore, mid-shelf and offshore regions, however, in this study the marine traps were installed within the inner GBR shelf to capture sediment resuspension events (dry season deployment) as well as the influence of SPM delivered through the riverine flood plumes (during the wet season deployments)”.

(Q): I think, the hypothesis that “the SPM in the GBR may not have terrestrial origin but produced locally in the marine environment” is justified by the data presented in this paper and could be valid for a typical day on GBR. On the other hand, during flood events, this hypothesis may not hold (at least, no data were presented in this paper to support it). Analogously, the conceptual model presented in this paper and describing the transformation of the SPM across the freshwater-marine environments may not be valid during such extreme weather events. I think, this paper will be improved by acknowledging such a limited scope of this hypothesis.

(R): In this study all the SPM samples were collected during the flood plume events within the wet season to capture pulsed inputs of SPM during flood events.

See above response (page14, lines 315-320).

(Q): Fig.2 captions – replace UEMZ with OEMZ

(R): UEMZ was changed to OEMZ.

(Q): References 2 and 3 have incomplete list of co-authors.

(R): References 2 and 3 have more than six authors. Please see below from nature communications guideline.

“*Nature Communications* uses standard Nature referencing style. All authors should be included in reference lists unless there are six or more, in which case only the first author should be given, followed by 'et al.’”